# Functional Genomics: From Soybean to Legume

**DOI:** 10.3390/ijms26136323

**Published:** 2025-06-30

**Authors:** Can Zhou, Haiyan Wang, Xiaobin Zhu, Yuqiu Li, Bo Zhang, Million Tadege, Shihao Wu, Zhaoming Qi, Zhengjun Xia

**Affiliations:** 1College of Agriculture, Northeast Agricultural University, Harbin 150030, China; canzhou0703@163.com (C.Z.); haiyanwang2001@163.com (H.W.); 2State Key Laboratory of Black Soils Conservation and Utilization, Northeast Institute of Geography and Agroecology, Chinese Academy of Sciences, Harbin 150081, China; ca_zhuxiaobin@163.com; 3Jilin Academy Agricultural of Sciences (Northeast Agricultural Research Center of China), Changchun 130033, China; lyqhb1@126.com; 4School of Plant and Environmental Sciences, Virginia Polytechnic Institute and State University, Blacksburg, VA 24061, USA; bozhang@vt.edu; 5Institute for Agricultural Biosciences, Oklahoma State University, Stillwater, OK 74078, USA; million.tadege@okstate.edu; 6Institute of Industrial Crops, Jiangsu Academy of Agricultural Sciences, Nanjing 210014, China; adidaswsh@yahoo.com

**Keywords:** legume, T2T, genome, pangenome, functional genomics, soybean

## Abstract

The Fabaceae family, the third-largest among flowering plants, is nutritionally vital, providing rich sources of protein, dietary fiber, vitamins, and minerals. Leguminous plants, such as soybeans, peas, and chickpeas, typically contain two to three times more protein than cereals like wheat and rice, with low fat content (primarily unsaturated fats) and no cholesterol, making them essential for cardiovascular health and blood sugar management. Since the release of the soybean genome in 2010, genomic research in Fabaceae has advanced dramatically. High-quality reference genomes have been assembled for key species, including soybeans (*Glycine max*), common beans (*Phaseolus vulgaris*), chickpeas (*Cicer arietinum*), and model legumes like *Medicago truncatula* and *Lotus japonicus*, leveraging long-read sequencing, single-cell technologies, and improved assembly algorithms. These advancements have enabled telomere-to-telomere (T2T) assemblies, pan-genome constructions, and the identification of structural variants (SVs) and presence/absence variations (PAVs), enriching our understanding of genetic diversity and domestication history. Functional genomic tools, such as CRISPR-Cas9 gene editing, mutagenesis, and high-throughput omics (transcriptomics, metabolomics), have elucidated regulatory networks controlling critical traits like photoperiod sensitivity (e.g., *E1* and *Tof16* genes in soybeans), seed development (*GmSWEET39* for oil/protein transport), nitrogen fixation efficiency, and stress resilience (e.g., *Rpp3* for rust resistance). Genome-wide association studies (GWAS) and comparative genomics have further linked genetic variants to agronomic traits, such as pod size in peanuts (*PSW1*) and flowering time in common beans (*COL2*). This review synthesizes recent breakthroughs in legume genomics, highlighting the integration of multi-omic approaches to accelerate gene cloning and functional confirmation of the genes cloned.

## 1. Leguminous Plants

Leguminous plants, commonly referred to as beans or pulses, belong to the Fabaceae family, the third-largest family of flowering plants after Orchidaceae (the orchid family) and Asteraceae (the aster or sunflower family) [1,2]. These species are widely distributed worldwide and play a vital role in agriculture, ecology, and human nutrition. The Fabaceae family is divided into several subfamilies, with Papilionoideae being particularly significant due to its abundance of edible crops [1,2].

Leguminous plants are rich in protein, dietary fiber, vitamins, and minerals, making them essential for a healthy diet [3]. Their seeds typically contain two to three times more protein than cereals such as wheat and rice, along with low fat content (primarily unsaturated fats) and no cholesterol [3,4]. These nutritional properties help reduce the risk of cardiovascular diseases and aid in blood sugar management.

Leguminous crops can be categorized into staple grains, oil crops, vegetables, forages, and miscellaneous grains. For example, soybeans and peanuts are primarily cultivated as staple grains or oil crops, whereas cowpeas, peas, and common beans are commonly consumed as vegetables. Other varieties, such as mung beans and chickpeas, are often used as side dishes or miscellaneous grains.

In agriculture, leguminous plants are frequently intercropped with other crops to enhance soil health and reduce pest and disease incidence. Their nitrogen-fixing ability improves soil fertility, benefiting subsequent crops. Current research and development efforts focus on increasing yield, improving nutritional content, enhancing nitrogen-fixing efficiency, and developing new varieties with greater resilience to environmental changes [5,6].

## 2. Genomic Advances in Leguminous Plants

Since the release of the soybean genome in 2010 [7], significant advancements have been made in the genomics of leguminous plants [8]. High-quality genome assemblies of species of agronomic or research importance, pan-genomes of wild and cultivated variants, single-cell atlases, and transcriptomic atlases have been developed (Table 1). Functional genomics in leguminous plants is also advancing rapidly, with efforts focused on understanding gene functions (Table 2) and regulatory networks through high-throughput sequencing, CRISPR/Cas9 editing [9], and multi-omic integration [10,11]. Here, we briefly review the current status of functional genomics in some major leguminous species, including soybeans, *Lotus japonicus*, *Medicago* spp., peas (*Pisum sativum*), peanuts (*Arachis hypogaea*), common beans (*Phaseolus vulgaris*), chickpeas (*Cicer arietinum*), cowpeas (*Vigna unguiculata*), pigeonpeas (*Cajanus cajan*), mung beans (*Vigna radiata*), adzuki beans (*Vigna angularis*), rice beans (*Vigna umbellata*), white lupin, *Mimosa*, *Dalbergia*, and licorice (*Glycyrrhiza* spp.). 2. Soybean

The soybean (*Glycine max*) is one of the most important legume crops globally. It is a versatile and nutritious crop that provides a significant source of protein, oil, and other essential nutrients. In addition to its agricultural importance, the soybean is also a model species for studying legume biology due to its relatively large genome size and complex genetic structure. The completion of high-quality telomere-to-telomere (T2T) genome assemblies, advancements in resequencing, and the development of pan-genomes have revolutionized our understanding of soybean’s genetic diversity, domestication history, and potential for crop improvement [12].

**Table 1 ijms-26-06323-t001:** List of some leguminous species having high-quality reference genome sequence.

Species	Cultivar or Accession	Genome Size (Mb)	Reference
Soybean (*Glycine max*)	Williams 82	1010	Schmutz et al. (2010) [7]; Espina et al. (2024) [13]; Wang et al. (2023) [14]
	Lee	1010	Valliyodan et al. (2020) [15]
	Zhonghuang 13 (ZH13)	1015–1025	Shen et al. (2018) [16]; Shen et al. (2019) [17]; Zhang et al. (2023) [18]; Zhang et al. (2024) [19]
	Jack	1012	Huang et al. (2023) [20]
	3 wild, 9 landraces, 14 cultivars	992.3–1059.8	Liu et al. (2020) [21]
	Nongdadou2	1010	Zhang et al. (2024) [22]
*Lotus japonicus*	Gifu	549	Sato et al. (2008) [23]
	Gifu	554	Kamal et al. (2020) [24]
*Medicago truncatula*	Mt3.5	375	Young et al. (2011) [25]
	Mt4.0	360	Tang et al. (2015) [26]
*Medicago sativa*	Zhongmu No.1	816	Shen et al. (2020) [27]
Pea (*Pisum sativum*)	Caméor	3920	Kreplak. et al. (2019) [28]
	ZW6	3719.6	Yang et al. (2022) [29]
	Zhewan No. 1	3930	Liu et al. (2024) [30]
Peanut (*Arachis hypogaea*)	Tifrunner	2540	Zhuang et al. (2019) [31]
	Yuanza 9102	2660	Zhang et al. (2024) [32]
Common Bean (*Phaseolus vulgaris*)	G19833	587	Schmutz et al. (2014) [33]
	Pinjin Yun 4 (PJY4)	560.61	Zhao et al. (2025) [34]
Chickpea (*Cicer arietinum*)	ICC 4958	740	Jain et al. (2013) [35]
	BGD218	895	Khan et al. (2024) [36]
Cowpea (*Vigna unguiculata*)	IT97K-499-35	519	Lonardi et al. (2019) [37]
	A147	539.4	Pan et al. (2023) [38]
	G98 (long-podded) and G323 (grain-type)	568.24 and 552.66	Wu et al. (2024) [39]
Pigeonpea (*Cajanus cajan*)	Asha	605.78	Varshney et al. (2012) [40]
	89 accessions	622	Zhao et al. (2020) [41]
Mung Beans (*Vigna radiata*)	Kamphaeng Saen 1	100.5	Tangphatsornruang et al. (2009) [42]
	JL7	475.19	Liu et al. (2022) [43]
	Weilü No. 9	500	Jia et al. (2024) [44]
Adzuki beans (*Vigna angularis*)	Jingnong6	489.8	Chu et al. (2024) [45]
Rice Bean (*Vigna umbellata*)	FF25 et al., 440 landraces	525.6	Guan et al. (2022) [46]
white lupin (*Lupinus albus*)	AMIGA	451	Hufnagel et al. (2020) [47]
*Mimosa bimucronata*	-	648	Jia et al. (2024) [48]
*Dalbergia odorifera*	-	653.45	Hong et al. (2020) [49]
*Glycyrrhiza uralensis*	308-19	379	Mochida et al. (2017) [50]

**Table 2 ijms-26-06323-t002:** The list of some important genes whose functions have been characterized in leguminous species.

Species	Gene Name	Function	Reference
Soybean	*E1*	Central regulator controlling photoperiodic flowering and maturity in soybeans	Xia et al. (2012) [51]
	*GmMDE*	Bridging *E1* and florigens GmFT2a/5a, represses flowering	Zhai et al. (2024) [52]
	*GmEID1*	Modulating light signaling through evening complex to control flowering time	Qin et al. (2023) [53]
	*E2*, *E2La*, *E2Lb*	Redundantly controlling photoperiodic flowering	Zhao et al. (2024) [54]
	*QNE1*	Key flowering regulator near the E1 locus	Xia et al. (2022) [55]
	*MS2/GmAMS1* (*Glyma.10G281800*)	Encoding a bHLH transcription factor and exhibiting tetrad-stage arrest and defective pollen wall development.	Fang et al. (2023) [56]
	*PH13*	Regulating plant height and shade tolerance	Qin et al. (2023) [57]
	*GmSW17*	Controlling seed width and weight	Liang et al. (2024) [58]
	*GmMs1* (*Glyma.13G114200*)	Encoding a kinesin-like protein essential for male fertility	Nadeem et al. (2021) [59]
	*barnase/barstar*	Inducing male sterility under the tapetum-specific *GmTA29* promoter by ablating the tapetum.	Szeluga et al. (2023) [60]
	*GmCOL1a*	Enhancing salt and drought tolerance in soybean	Xu et al. (2023) [61]
	*Rpp3*	Conferring resistance to soybean rust caused by *Phakopsora pachyrhizi*,	Bish et al., (2024) [62]
*Lotus japonicus*	*LjNLP3*	Promoting nodules transition from early to mature stages	Ye et al. (2024) [63]
	*IAMT1*	Promoting root nodule development	Goto et al. (2022) [64]
	*LjNRT2.1*	Essential for nitrate-mediated suppression of nodule formation	Misawa et al. (2022) [65]
*Medicago truncatula*	*MtLICK1/2*	Balancing rhizobial symbiosis and plant immunity	Wang et al. (2025) [66]
	*PINNA2*	Controlling compound leaf morphogenesis	He et al. (2024) [67]
*Medicago sativa*	*HDL*	Increasing biomass and delaying flowering in alfalfa	Wang et al. (2025) [68]
Pea	*chlorophyll synthase* (*ChlG*)	Explaining the parchmentless phenotype, leading yellow pod	Feng et al. (2025) [69]
Peanut	*AhFAX1*	Controlling seed size	Liu et al. (2022) [70]
	*AhDPB2*	Controlling seed length	Liu et al. (2022) [70]
	*PSC1*	Controlling testa color	Zhao et al. (2025) [71]
	*AhSAMS1*	Enhancing salt tolerance through Ca^2+^/CaM signaling	Yang et al. (2025) [72]
Common Bean	*PvE1*	Repressing flowering	Zhang et al. 2016 [73]
	*CONSTANS-like* *(COL2)*	Promoting flowering	González et al. 2021 [74]
	*PvPW1*	Associating with pod width	Xu et al. (2024) [75]
	*MS-2*	Leading to male sterility due to a splice site mutation	Xu et al. (2023) [76]
	*PvFtsH2*	Regulating the degradation of photodamaged PSII RC D1 protein	Xu et al. (2021) [77]
Chickpea	*MPL1*	Regulating leaflet initiation	Liu et al. (2023) [78]
Chickpea	*CaABCC3*	Regulating seed size and weight	Basu et al. (2019) [79]
Chickpea	*Ca-miR164e*	Targeting CaNAC100, thus affecting seed protein content and weight	Chakraborty et al. (2024) [80]

### 2.1. Genome Sequencing and Assembly

Since the first release of the soybean genome sequence in 2010 [7], the resequencing of cultivars, accessions, and genetic populations has greatly facilitated the identification of polymorphic markers, such as single nucleotide polymorphisms (SNPs) and structural variants (SVs). These advancements have subsequently advanced genome-wide association studies (GWASs), mapping, and gene cloning. Meanwhile, the assembly of several other cultivars (Table 1), such as Zhonghuang 13 (ZH13) [16,17] and Lee [15], has been achieved. Initial efforts focused on developing highly contiguous, nearly gapless genome assemblies for two economically important soybean cultivars, *Williams 82* and *Lee*, using complementary genomic technologies and assembly algorithms. A total of 10 chromosomes in Williams 82 and 8 in Lee were entirely reconstructed into single contigs without any gap [15]. These assemblies identified 58,287 and 56,725 protein-coding genes in Williams 82 and Lee, respectively.

The quality of the soybean genome has been significantly improved, especially with the recent establishment of over a dozen high T2T quality genome assemblies [13,14,18,19,20,21,81]. An updated assembly (version Wm82.a6) of the soybean cultivar *Williams 82* was provided, derived from a specific sub-line known as Wm82-ISU-01 [13]. The genome was assembled using high-fidelity (HiFi) long reads from PacBio (PacBio HiFi reads) and integrated into chromosomes using High-throughput Chromosome Conformation Capture (Hi-C) data [13]. This assembly revealed substantial genomic heterogeneity, including a large heterozygous region on chromosome 12. The 20 soybean chromosomes were assembled into a genome of 1.01 Gb, consisting of 36 contigs, with a total of 48,387 gene models identified. Another T2T assembly of *Williams 82* was also released [14]. Additionally, the genome of ‘*Fiskeby III*’ was assembled and compared with Wm82.a6 to reveal nucleotide and structural polymorphisms within a QTL region for iron deficiency chlorosis resistance [13].

Recently, two complementary high-quality genome assemblies of *Zhonghuang 13* have been released [18,19]. The first T2T assembly of *Zhonghuang 13* successfully fills all 393 previously unsequenced gaps, yielding a genome length of 1,015,024,879 bp and an N50 length of 52,033,905 bp [19]. Notably, all 40 telomeric regions across the 20 chromosomes were completely assembled, with a median telomere length of 8449 bp. The assembly quality is further validated by a BUSCO score of 99.8% and a base quality score (Merqury quality score) of 46.441. A total of 50,564 high-confidence protein-coding genes were annotated, including 707 novel genes [19]. In contrast, another version of Zhonghuang 13 was assembled using a combination of ultra-long reads from Oxford Nanopore Technology (ONT reads), PacBio HiFi reads, and short reads from BGI Genomics [18]. This assembly contains nearly complete telomeres (39 out of 40) and features an enriched “CCCTAAA” specific repeat as well as complete assembled centromeres containing *Cent-Gm1* or *Cent-Gm2*. The total genome size is 1,007,237,669 bp, with a contig N50 of 48.76 Mb [18]. While this assembly also achieves high completeness in telomeric regions, it falls slightly short in the number of fully assembled telomeres compared to the first T2T assembly. These complementary features highlight the strengths of different assembly strategies and underscore the importance of multiple approaches in achieving comprehensive genome assemblies.

The complete genome assembly of the soybean cultivar Jack was obtained, resulting in a total length of 1,011,764,152 bp [20]. A total of 63,703 genes were annotated in the Jack genome, with 16,914 considered to be newly annotated compared to the published Wm82.a4.v1 annotation [20]. The assembly-based comparative method “Asm2sv” identified SVs comprehensively, enabling pan-genome analysis of 462 worldwide cultivars and varieties [81]. Selective sweeps between Japanese and US soybeans were identified, including the pod-shattering resistance gene *PDH1* [81]. Genome-wide association studies further identified several quantitative trait loci associated with large-seed phenotypes in Japanese soybean lines [81].

Liu et al. (2020) used 26 soybean accessions (3 wild, 9 landraces, 14 cultivars) to construct a pan-genome with 57,492 gene families and identified 723,862 presence/absence variants (PAVs), 27,531 copy number variants (CNVs), and other structural variations [21]. The PanSoy pan-genome was constructed using 204 cultivated soybean varieties, identifying 108 Mb of new non-reference sequences containing 3621 protein-coding genes with a high core gene content (90.6%) and significant enrichment of variable genes in resistance and signaling pathways [82]. A high-quality genome of Nongdadou2 (NDD2) were assembled, in which 25,814 SV-gene pairs were identified [22]. Thirteen NDD2-specific SVs were validated via deep resequencing of 547 accessions. SVs related to seed protein, weight, drought adaptation, and yield/seed quality traits were found, with 12 affecting oil and isoflavone contents [22].

Single-cell genomics has provided unprecedented insights into the molecular mechanisms of soybean development and gene regulation. A single-cell atlas with 303,199 chromatin accessible regions and 103 cell types was generated [83]. Key transcription factors and regulatory networks in nodule development and seed embryogenesis were identified, offering insights into the molecular mechanisms of soybean development at the single-cell level. Additionally, a comprehensive transcriptomic atlas covering 314 samples from various organs greatly benefits the identification of key genes and regulatory networks in nodule formation, leaf photosynthesis, and root development [84].

Based on accessible chromatin region (ACR) datasets developed from leaf, flower, and seed tissues using MNase-hypersensitivity sequencing, the gain or loss of flanking ACRs and mutation of cis-regulatory elements (CREs) within the ACRs can alter the balance of the expression level and/or tissue specificity of the duplicated genes [85]. Photosynthesis and nitrogen fixation can be affected by genome-wide non-CG DNA hypomethylation [86].

The knockout of *GmMET1s* by CRISPR-Cas9, along with comprehensive methylome and transcriptome analyses, reveals the well-coordinated regulatory impact of mCG on soybean gene expression, which can influence soybean growth, development, and stress responses [87].

### 2.2. Functional Genomics and Gene Function

In recent years, significant progress has been made in various aspects of soybean research, including photoperiod response, nodule formation and regulation, male sterility, agronomic traits related to soybean yield such as plant height and lodging resistance, as well as resistance to biotic and abiotic stresses (Table 2).

### 2.3. Plant Growth and Development

Since the successful deciphering of the molecular identity of *E1* in 2012 [51], the functional network of photoperiodic flowering regulation in soybeans has been greatly advanced [52]. Soybean mutants carrying edited nuclear localization signals (NLSs) of *E1* and *E1Lb* can reduce photoperiod sensitivity [88]. These mutants showed shortened flowering and maturity times, making them suitable for temperate regions [88]. *E1* gene duplicates underwent subfunctionalization, with *E1La* and *E1Lb* showing reduced nuclear localization and altered regulation of flowering time genes [51,89]. The *Tof4* (Time of flowering 4) locus, encoded by an *E1* homolog, *E1La*, represses flowering and enhances adaptation to high latitudes in wild soybeans [90]. *GmMDE* genes bridge the maturity gene *E1* and florigens *GmFT2a/5a*, while *E1* directly binds to *GmMDE* promoters, increasing H3K27me3 levels and repressing their expression [52]. *GmEID1* modulates light signaling through the evening complex (EC) to control flowering time. Targeted mutations in *GmEID1* improved yield across different latitudes [53]. *E2*, *E2La*, and *E2Lb* redundantly control photoperiodic flowering in soybeans. A feedback loop where *E2* interacts with *J/ELF3* to form a complex that degrades *J* protein, relieving the inhibition of *E1* expression, was discovered [54].

*qFT13-3*, which was cloned as a PRR gene, functions as a flowering inhibitor by directly downregulating the expression of *GmELF3b-2*, a component of the circadian clock evening complex [91]. *Flowering 16* (*Tof16*), an *LHY* homolog, functions as a flowering repressor and forms the major genetic basis of soybean adaptation into the tropics along with *J* [92]. *QNE1* (*Glyma.06G23400*) functions as a key flowering regulator near the *E1* locus. Non-synonymous mutations in *QNE1* are associated with flowering time, providing a basis for breeding widely adaptable soybean varieties [55]. *Glyma.08G293800*, encoding proline-rich protein 4-like, was associated with sucrose content by both Protein Variation Effect Analyzer (PROVEAN) and RNA-seq analyses [93].

High-throughput sequencing (STRIPE-seq) was employed to survey transcription initiation at promoter elements (TSSs). Duplicated genes possessed more TSRs (TSS clusters/regions), exhibited lower degrees of tissue specificity, and underwent stronger purifying selection than singletons [94].

Through genome editing, the loss-of-function of *miR396* genes can lead to significantly enlarged seed size in soybeans [95].

In soybeans, CRISPR-Cas9 was used to knock out *GmDCL2a* and *GmDCL2b*, revealing that most 22-nucleotide siRNAs originate from LIRs formed by *CHS1* and *CHS3* genes [96]. These siRNAs target other *CHS* genes and induce secondary 21-nucleotide siRNAs. Disruption in mutants increased *CHS* mRNA levels, changing seed coat color from yellow to brown, highlighting *GmDCL2*’s role in processing LIR-derived transcripts and regulating plant gene expression [96].

The gene *GmSWEET39*, which is primarily detected in the parenchyma and integument tissues of the seed coat, influences the accumulation of oil and protein within the developing embryo by modulating the transport of sugars from the maternal seed coat to the filial embryo [97].

*PH13* was proven to be a key regulator of plant height and shade tolerance. The *PH13-H3* haplotype, containing a *Ty1/Copia*-like retrotransposon insertion, results in a truncated protein that weakens interaction with *GmCOP1s*, leading to increased *STF1/2* accumulation and reduced plant height [57]. *GmSW17*, a ubiquitin-specific protease, was identified as a major QTL controlling seed width and weight. *GmSW17* interacts with *GmSGF11* and *GmENY2* to form a deubiquitinase module, regulating *H2Bub* levels and inhibiting the G1-to-S transition [58].

*GmNNL1*, encoding an R protein, was proven to directly interact with *NopP* effector from *Bradyrhizobium USDA110* to trigger immunity and inhibit nodulation through root hair infection [98].

The *cle1a/2a* (*ric1a/2a*) mutant, with moderately increased nodules, showed enhanced biomass and yield, leading to a 10–20% increase in yield and 1–2% increase in protein content in the field [99]. Transcription factors *GmWRI1a/b* and *GmZF351* were found to be associated with the regulation of oil content in soybeans [100]. The molecular identities for *Pd1* (dense pubescence), *Ps* (sparse pubescence), and *P1* (glabrous) were identified through GWS and map-based cloning. *Pd1*, *Ps*, and *P1* function as a complex feedback loop to regulate pubescence formation in soybeans [101].

Using forward genetic methods and CRISPR/Cas9 gene editing, *GmKIX8-1* encoding KIX domain-containing proteins, was proven to regulate organ size, thus changing seed weight [102].

Male sterility in soybeans is a critical trait for hybrid breeding, offering a pathway to harness heterosis and improve yield. Recent advances in genomics have enabled the identification and functional characterization of key male sterility genes, providing insights into their roles in reproductive development.

The *GmMs1* (*Glyma.13G114200*) gene, cloned from the *ms1* mutant, encodes a kinesin-like protein essential for male fertility. It is involved in meiotic cytokinesis and pollen development. Mutations in *GmMs1* lead to abnormal pollen wall formation and tapetal degradation, resulting in complete male sterility. Metabolic profiling revealed disruptions in starch, sucrose, and flavonoid pathways in sterile anthers [59]. The *MS2/GmAMS1* (*Glyma.10G281800*), identified in the *ms2* mutant, encodes a bHLH transcription factor homologous to *Arabidopsis AMS*. It regulates genes involved in secondary metabolite biosynthesis and lipid metabolism, crucial for microspore cell wall formation. The *ms2* mutant exhibits tetrad-stage arrest and defective pollen wall development. Notably, *ms2* maintains high female fertility, making it ideal for hybrid breeding [56].

A biotechnological approach introduced the barnase/barstar system to induce male sterility. Barnase, expressed under the tapetum-specific *GmTA29* promoter, ablates the tapetum, causing sterility. Fertility is restored in F1 hybrids by crossing with barstar-expressing lines, which inhibit barnase. This system demonstrates the importance of dosage balance for successful fertility rescue [60].

### 2.4. Resistance to Abiotic and Biotic Stress

*GmCOL1a*, a *CONSTANS*-like gene, enhances salt and drought tolerance in soybeans [61]. The overexpression of *GmCOL1a* increased leaf relative water content and proline content and reduced MDA and ROS levels [61].

The *Rpp3* locus, which confers resistance to soybean rust caused by *Phakopsora pachyrhizi*, was fine-mapped to a 371 kb region on chromosome 6 [62]. Five *Rpp3* candidate (*Rpp3C*) NBS-LRR genes were identified in resistant soybean lines, and co-silencing these genes compromised resistance to *P. pachyrhizi* [62]. Gene expression analysis and sequence comparisons suggest that a single candidate gene, *Rpp3C3*, is responsible for *Rpp3*-mediated resistance [62].

CRISPR/Cas9-mediated editing of *GmTAP1* confers enhanced resistance to *Phytophthora sojae*, a major threat to soybean production worldwide [103].

The combination of transcriptomes and a GWS analysis revealed that a 7 bp InDel in the promoter of *GsERD15B* is significantly associated with the salt tolerance of soybeans. The overexpression of *GsERD15B* enhanced salt tolerance possibly via moderating ABA-signaling, proline content, catalase peroxidase, dehydration response, and cation transport [104].

## 3. *Lotus japonicus*

Native to Asia, *Lotus japonicus* is a small, perennial legume with trifoliate leaves. It is cultivated both as a forage crop and for its ornamental appeal. Due to its relatively small genome size of about 500 Mb, ease of genetic transformation, and short growth cycle (approximately 3–4 months), *Lotus japonicus* is regarded as a model species for studying legume biology and symbiotic relationships.

### 3.1. Genome Sequencing and Assembly

The first draft genome sequence of *Lotus japonicus* was published in 2008, covering approximately 91.6% of the euchromatic regions of the genome and identifying 36,488 protein-coding genes [23]. In 2016, all available Lotus data was publicly available at *Lotus* Base, including mutant populations, reference genome, annotated proteins, and expression profiling data [105]. In 2020, the *L. japonicus* reference genome sequence was further improved using long PacBio reads from the Gifu accession, combined with Hi-C data and new high-density genetic maps [24]. This effort resulted in a set of six chromosome-scale scaffolds containing 549 Mb of sequence, representing over 99% of the draft assembly. The genome of *Lotus japonicus* shares significant synteny with other legumes, such as *Medicago truncatula* and *Glycine max*, allowing for the identification of conserved and divergent genomic regions [23,106]. A whole-genome duplication (WGD) event, which predates the divergence of *Lotus japonicus* and *Medicago truncatula*, has played a crucial role in shaping the genome architecture and contributing to the expansion of gene families associated with nitrogen fixation and symbiosis [23,106].

### 3.2. Functional Genomics and Gene Function

*In general*, auxin-related genes are preferentially expressed in the pulvinus and have been reported to be involved in the regulation of nyctinastic leaf movement. Transcriptome and qRT-PCR analyses revealed that at least two auxin-related genes, *IAA19/FLS1*, are dominantly expressed in the pulvinus [107].

A comparative transcriptomic analysis revealed that tissue-specific regulation of lipid polyester synthesis genes, such as *Fatty acyl-CoA Reductase*, can mediate oxygen permeation into nodules. This affects the deposition of polyesters on this cell layer and promotes nodule permeability [108].

Stereo-seq spatial transcriptomics was used to map the spatiotemporal transcriptomic landscape of nodule organogenesis in *Lotus japonicus*. *LjNLP3*, a NIN-LIKE PROTEIN (NLP) family member, functions in transitioning nodules from early to mature developmental stages [63].

The role of the indole-3-acetic acid (IAA) carboxyl methyltransferase gene *IAMT1* in promoting root nodule development in *Lotus japonicus* was identified [64]. *IAMT1* expression was found to be crucial for cortical cell division during nodule formation, and methyl-IAA (MeIAA) was identified as inducing the expression of the key nodule development transcription factor *NIN* [64].

*LjNRT2.1* was identified as essential for the nitrate-mediated suppression of nodule formation and functions in the same genetic pathway as *LjNLP1* and *LjNLP4* [65].

The Casparian strip was identified as regulating nodule formation in *Lotus japonicus* by modulating nutrient exchange and signaling pathways, including the AON system and *miR2111*. Mutants of genes homologous to those regulating Casparian strip formation in *Arabidopsis*, such as *Ljmyb36*, *Ljsgn3*, and *Ljsgn1*, exhibited delayed nodule formation, reduced nodule numbers, and impaired nitrogen fixation [109].

## 4. *Medicago*

The genus *Medicago*, commonly known as alfalfa or lucerne, comprises hardy legumes characterized by clover-like leaves and small flowers. These plants are highly nutritious, are rich in protein and vitamins, and are widely used as fodder for livestock and as cover crops to enrich soil through nitrogen fixation. Their deep root systems also contribute to soil structure improvement.

### 4.1. Genome Sequencing and Assembly

In 2011, the draft sequence of the *Medicago truncatula* euchromatin was released [25], capturing approximately 94% of all genes through a combination of BAC assembly and Illumina shotgun sequencing. This draft sequence highlighted the role of a whole-genome duplication (WGD) event approximately 58 million years ago in shaping the *M. truncatula* genome and contributing to the evolution of endosymbiotic nitrogen fixation [25].

An improved version (Mt4.0) of the *M. truncatula* genome was released in 2014 [26]. The Mt4.0 assembly encompassed about 360 Mb of actual sequences. The re-annotation process integrated EST, RNA-seq, protein, and gene prediction evidence, resulting in a comprehensive gene set of 50,894 genes [26]. A high-quality haploid genome of 3.11 Gb was assembled, with a contig N50 of 5.67 Mb and scaffold N50 of 97.41 Mb [110]. In this study, 97.44% of the assembly was anchored to 32 pseudochromosomes, and 138,037 protein-coding genes with an average length of 3525.54 bp were identified. In the comparison of *Medicago truncatula* and *Medicago sativa*, significant structural variations (SVs) were found, including translocations and inversions [110]. Additionally, a chromosome-level genome of wild diploid alfalfa (*Medicago sativa* ssp. *falcata*) was assembled, with a size of 3.11 Gb and containing 51,936 protein-coding genes. The accumulation of LTR transposable elements (TEs) is regarded as a major factor driving the genome expansion of cultivated alfalfa [111].

The pan-genome of *M. truncatula*, constructed from 15 accessions, includes core and dispensable sequences, with structural variants and transposable elements identified in gene families such as NBS-LRRs [112]. A de novo assembled 816 Mb high-quality, chromosome-level haploid genome sequence of ‘Zhongmu No.1’ alfalfa, a heterozygous autotetraploid, was reported [27]. A genomic population analysis of 162 alfalfa accessions revealed high genetic diversity, weak population structure, and extensive gene flow from wild to cultivated alfalfa [27]. Based on the resequencing analysis of 220 *Medicago sativa* accessions, four subpopulations (Q1-Q4) with distinct genetic structures were identified [113].

The genome assembly of accession SA27063 identified promising candidates for spring black stem and leaf spot disease resistance, such as a 10.85 kbp retrotransposon-like insertion disrupting a ubiquitin conjugating E2 [114].

### 4.2. Functional Genomics and Gene Function

The GOLVEN peptides modulate root morphology and nodule ontogeny by suppressing the expression of microtubule-related genes and exerting their effects by changing the expression of a large subset of auxin-responsive genes based on a transcriptome analysis [115].

Using integrative omic analyses coupled with molecular approaches, *MtPIN4* was proven to play critical roles in amino acid biosynthesis and metabolism of seeds, thus affecting seed size [116]. The role of the GRAS transcription factor *PINNA2* in controlling compound leaf morphogenesis in *Medicago truncatula* was clarified [67]. *PINNA2* directly binds to the promoter of *SINGLE LEAFLET1* (*SGL1*) and downregulates its expression, which is crucial for leaflet initiation. *PINNA2* interacts with other repressors, *PINNA1* and *PALM1*, to precisely define the spatiotemporal expression of *SGL1* in compound leaf primordia [67]. The overexpression of the WUSCHEL orthologue *HEADLESS* (*HDL*) in the salt-tolerant alfalfa variety *Zhongmu No. 1* increased biomass, crude protein, neutral detergent fiber, and micronutrient content [68]. *HDL* also delays flowering by inhibiting the expression of the flowering time gene *MsFTa1* and promotes branching by inhibiting the expression of the strigolactone synthesis gene *MsMAX3* [68].

In an assessment of six root system architecture (RSA) traits in 171 alfalfa genotypes under normal and drought conditions, 60 significant SNPs and 19 high-priority candidate genes associated with RSA traits were identified [117]. A total of 17 genomic regions under selection were significantly associated with important agronomic traits, including root fresh weight and the root-to-dry weight ratio. Candidate genes such as *MsTIR* and *MsPGL3* are regarded as being associated with root development [113].

A novel kinase, *MtLICK1/2*, is proven to play a crucial role in balancing rhizobial symbiosis and plant immunity in *Medicago truncatula* [66]. The researchers identified that *MtLICK1/2* interacts with the nodule factor receptor *MtLYK3*, facilitating symbiotic signaling while suppressing plant immune responses. This interaction is vital for rhizobial infection and nodule development. *MtLICK1/2* and *MtLYK3* undergo reciprocal trans-phosphorylation, activating downstream signaling pathways necessary for symbiosis [66]. *MtLICK1/2* suppresses plant immunity triggered by rhizobia, allowing for successful symbiosis without compromising the plant’s defense mechanisms [66].

After analyzing 704 *Medicago* accessions representing 24 species to identify genes related to climate adaptation, 1671 candidate genes associated with climate adaptation, including *CYP450* and *DREB*, were identified [118]. Genome-wide identification and a transcriptome analysis have revealed the functions of an alfalfa MYB-like transcription factor *MsMYBH* on drought resistance via *MsWAV3*-mediated degradation [119]. The *LEA* gene is involved in the response to drought, salinity, and cold stress [120]. The U-Box gene family is involved in responses to abiotic stresses [121].

## 5. Pea (*Pisum sativum*)

The pea (*Pisum sativum*) is not only of significant agronomic importance but also an ideal model organism for genetic studies due to its ease of cultivation and short life cycle. These characteristics led Gregor Mendel [122] to select peas as his model organism for genetic research, ultimately leading to the formulation of the fundamental laws of inheritance, such as the law of segregation and the law of independent assortment.

### Genome Sequencing and Assembly

The first annotated chromosome-level reference genome (v.1a) for peas was reported in 2019 [28]. This assembly comprises 3.92 Gb, representing approximately 88% of the estimated pea genome size (~4.45 Gb), with 82.5% (3.23 Gb) of the sequences assigned to the seven pseudomolecules (chromosomes). The pea genome exhibits intense gene dynamics and significant expansion in genome size, likely associated with the divergence of the Fabeae tribe from its sister tribes [28]. In 2022, a further improved pea reference genome was released, achieving a 243-fold increase in contig length and significant enhancements in sequence quality, particularly in complex repeat regions [29].

A pan-genome established from 118 cultivated and wild peas enabled us to reveal that *Pisum abyssinicum* is a separate species different from *P. fulvum* and *P. sativum* within *Pisum* [29]. It also identified two known Mendel’s genes related to stem length (*Le/le*) and seed shape (*R/r*) that were studied by Mendel [29,122]. Furthermore, a high-quality chromosome-level genome of ‘Zhewan No. 1’ was assembled, with a genome size of 3.93 Gb and a contig N50 of 4.23 Mb [30]. This assembly was assigned to seven chromosomes, with a scaffold N50 of 533.44 Mb, in which a total of 43,957 protein-coding genes were annotated [30]. Based on a comprehensive map of genetic variation yielded through resequencing data from 314 accessions, 235 candidate loci associated with 57 important agronomic traits, especially stem length (*Le/le*), flower color (*A/a*), cotyledon color (*I/i*), and seed shape (*R/r*), were identified through genome-wide association studies [30]. The first pan-plastome genome map of peas was generated, revealing structural features and genetic diversity [123]. High variability was identified in genes such as *ycf1*, *rpoC2*, and *matK*, which are suitable for DNA barcoding. Two independent domestication events in cultivated peas were discovered, providing insights into the evolutionary history of peas through a phylogenetic analysis [123]. Leveraging the newly released Pea Single Plant Plus Collection (PSPPC) and reference genome, 54,344 high-quality SNPs were used to conduct the first genome-wide association study (GWAS) for mineral traits in peas, revealing significant SNPs and candidate genes were identified for iron, phosphorus, and zinc, providing foundational genomic targets for biofortification breeding [124].

Boutet et al. (2016) demonstrated the potential for large-scale SNP discovery and genetic mapping in peas using whole-genome sequencing and genotyping-by-sequencing (GBS) [125]. They identified over 130,000 high-quality SNPs and constructed a genetic map collinear with the *Medicago truncatula* genome, highlighting the potential for marker-assisted breeding [125]. Tayeh et al. (2015) developed the GenoPea 13.2K SNP Array and a high-density, high-resolution consensus genetic map [126].

In the 1860s, Mendel proposed the theory that “hereditary factors” control traits for the first time through his hybridization experiments on seven pairs of contrasting traits in peas (seed shape, cotyledon color, flower color, flower position, pod shape, pod color, and plant height) and deduced the laws of independent assortment and free combination of genetic variations across generations [122]. However, the key genes controlling pod color, pod shape, and flower position remained mysteries in genetics for more than a century.

The genetic basis of 72 key agronomic traits has been recently identified, completing the last piece of the puzzle of Mendel’s seven classic genetic traits in peas [69]. New alleles for four characterized Mendelian genes (R, Le, I, A) and genes underlying three uncharacterized traits (Gp, P, Fa) were revealed by resequencing 697 accessions from global germplasm collections and GWASs [69]. A 100 kb deletion upstream of the chlorophyll synthase (ChlG) gene was found to cause yellow pods in gp mutants. A premature stop codon in Dodeca-CLE41/44 was identified, explaining the parchmentless phenotype [69]. Significant changes in gene expression related to seed development, dormancy, and defense mechanisms were identified [127]. Alterations in secondary metabolites, such as phenolic compounds, were also identified, which may affect seed dormancy and germination [127].

Genome-wide identification and comparative mapping were conducted to disclose the function of *SPL* genes on abiotic stress [128], *WRKY* genes [129], and *TCP* genes in response to salt stress [130]. *Lathyrus MLO1* belongs to Clade V, like all dicot *MLO* proteins associated with powdery mildew susceptibility [131]. Nuclear *POLLUX* ion channels cofunction with Ca^2+^ channels to generate Ca^2+^ signals, critical for establishing mycorrhizal symbiosis and root development [132].

The importance of the major photoperiod sensitivity locus *Hr/PsELF3a* and the identification of two other loci on chromosomes 1 (*DTF1*) and 3 (*DTF3*) that contribute to earlier flowering in the domesticated line under both photoperiods were highlighted [133].

## 6. Peanut (*Arachis hypogaea*)

Peanuts (*Arachis hypogaea*), native to South America, are unique legumes that grow underground and are known for their nutty flavor and rich nutritional profile, including high levels of protein, healthy fats, and vitamins. It is an important crop for both food and oil production.

### 6.1. Genome Sequencing and Assembly

Due to the allotetraploid nature of cultivated peanuts (*Arachis hypogaea*) with a genome size of about ∼2.7 Gb, the diploid progenitors of peanuts, *Arachis duranensis* and *Arachis ipaensis*, were sequenced first in 2016. These genomes were proven to be similar to the cultivated peanut’s A and B subgenomes and were used to identify candidate disease resistance genes, guide tetraploid transcript assemblies, and detect genetic exchange between the cultivated peanut’s subgenomes, providing insights into the ancestral origins of the cultivated peanut [134]. In 2019, a comprehensive genome assembly of peanuts was released, providing a detailed map of the tetraploid genome [31]. This assembly includes 20 pseudomolecules and 83,709 protein-coding gene models, covering approximately 2.54 Gb of the genome [31].

A telomere-to-telomere (T2T) genome assembly of the cultivated peanut variety “Yuanza 9102” was generated, adding 123.2 Mb of new sequences containing 1672 protein-coding genes compared to the previous Tifrunner reference genome [32]. Both the A and B subgenomes were found to have expanded around 4000 years ago, coinciding with the earliest archaeological evidence of peanut cultivation. This expansion was primarily due to slower repeat sequence removal, rather than more insertions, leading to higher repeat content in the centromeric regions of subgenome B [135].

Re-sequencing of 203 accessions enabled their classification into four genetic groups [70]. Key genes under selection during peanut domestication and improvement were proposed, including those related to auxin, plant architecture, nitrogen utilization, and seed size. *AhFAX1* is a key candidate gene controlling seed size, with a non-synonymous SNP in its sixth exon causing a cysteine-to-tyrosine substitution, associated with increased seed weight. *AhDPB2* was identified as a candidate gene for seed length, encoding a key protein in DNA replication and cell cycle progression [70]. High-density genetic maps have been constructed using single nucleotide polymorphisms (SNPs) and other markers [136].

A single-cell gene expression atlas of peanut stem tissues was released, identifying 29,308 stem cells and 53,349 gene expression profiles [137]. A total of 2053 cell type-specific upregulated genes and 3306 genes involved in cell development and differentiation were identified [137].

A comprehensive predicted protein–protein interaction network yielded through homologous mapping can be used to identify candidate genes and validate functional interactions [138].

### 6.2. Functional Genomics and Gene Function

Genes involved in oil biosynthesis, such as those encoding enzymes in the fatty acid synthesis pathway, have been identified to improve oil content and quality in peanuts [139].

The gene controlling testa color was mapped to a 3.61 Mb interval on chromosome 03, identifying a bHLH transcription factor gene, *PSC1*, with a 35 bp insertion in its promoter in red testa varieties [71]. *PSC1* forms a complex with *AhMYB7* to inhibit the expression of *ANR* (anthocyanidin reductase), increasing anthocyanin accumulation and deepening testa color. The function of *PSC1* was confirmed through genetic transformation in *Arabidopsis* and peanut calli [71]. At first, molecular markers were developed and used in marker-assisted selection (MAS) programs to accelerate the breeding of improved peanut varieties [140]. High-throughput sequencing technologies have been used to analyze genetic diversity in peanut populations [141]; genome-wide association studies revealed genetic loci and candidate genes for pod-related traits in peanuts [142] and luteolin content on peanut shells [143]. Particularly, *PSW1*, an LRR receptor kinase, controls pod size through the upregulation of *PSW1(HapII)* (super-large pod allele of *PSW1*), a positive regulator of pod stemness *PLETHORA 1 (PLT1)*, thereby resulting in a larger pod size [144].

Chromatin spatial organization of wild type and mutant peanuts reveals high-resolution genomic architecture and interaction alterations [145]. About 2.0% of chromosomal regions switch from inactive to active in the mutant line when compared with the WT type, harboring 58 differentially expressed genes enriched in flavonoid biosynthesis and circadian rhythm functions. Particularly, a specific upstream AP2EREBP-binding motif might upregulate the expression of the *GA2ox* gene and decrease active gibberellin (GA) content, presumably making the mutant plant dwarf [145].

Huai et al. (2024) used CRISPR/Cas9 gene editing to knock out *AhKCS1* and *AhKCS28* genes, reducing VLCFA (very long chain fatty acids) content to 0.4–0.9% in seeds [146]. The double mutants had no significant impact on peanut growth and development [146].

Advances in genetic transformation have enabled the development of transgenic peanuts with improved traits, such as increased fragrance [147].

*AhWRKY70*, a WRKY family transcription factor, was found to be highly expressed in the cortex and xylem of stems and promotes stem development through auxin and ethylene pathways. *AhWRKY70* expression is positively correlated with stem height and is regulated by an SNP in its promoter region [137].

Three hundred eighty-one de novo genes derived from non-coding sequences in *A. hypogaea* cv. Tifrunner based on comparison with five closely related *Arachis* species were systematically identified, which may function pluripotently in responses to biotic stresses as well as in growth and development [148].

Additionally, genes related to disease resistance have been identified through comparative genomics and functional analysis. For example, resistance genes against fungal pathogens have been discovered [149]. Genes involved in drought tolerance have also been identified through transcriptomic and genomic analyses [150].

The expression of *AhSAMS1* (S-adenosylmethionine synthase 1) was shown to be induced by Ca^2+^, ABA, and salt stress. The overexpression of *AhSAMS1* increased polyamine synthesis, reduced ethylene levels, and scavenged ROS, thereby enhancing salt tolerance [72]. Detailed studies on this new CaM4-binding protein demonstrated that *AhSAMS1* regulated ion homeostasis and improved salt tolerance through Ca^2+^/CaM signaling [72].

Genome-wide identification and a transcriptome analysis were conducted to disclose the function of Heavy Metal ATPase (HMA) on heavy metal transport [151], sugar invertase genes on abiotic stress [152], microRNAs on drought and heat stress [153], the type III effector RipBB on *Ralstonia solanacearum*—peanut interaction [154], and ANK-containing effector functions in host cells, helping to understand the mechanism of *R. pseudosolanacearum*—peanut interaction [154]. A review summarized new progress in identification of resistance loci to pest and disease and their incorporation into peanut using marker-assisted selection (MAS) and genomic tools [155].

## 7. Common Bean (*Phaseolus vulgaris*)

Common beans (*Phaseolus vulgaris*) are versatile legumes known for their diverse shapes and colors, including kidney beans, pinto beans, and navy beans. Their richness in protein, fiber, and nutrients makes them beneficial for heart health and digestion. The ease of cultivation and adaptability to various climates have resulted in their widespread cultivation worldwide.

### 7.1. Genome Sequencing and Assembly

The first reference genome for common beans was released in 2014, assembling 473 Mb of the 587-Mb genome and anchoring 98% of this sequence into 11 chromosome-scale pseudomolecules [33]. This foundational work revealed the genetic basis of domestication and identified genes linked to traits such as increased leaf and seed size [33].

In 2025, a gap-free telomere-to-telomere (T2T) genome of 560.61 Mb was assembled using the red kidney bean variety “Pinjin Yun 4” (PJY4) [34]. Two whole-genome duplication events and expanded gene families related to phenylpropanoid metabolism and ABC transporters were identified. Key metabolic pathways and differentially expressed metabolites related to seed coat pigmentation were identified through a metabolomic analysis [34].

A pan-genome analysis of common beans was conducted in 2024, revealing extensive gene loss during domestication and range expansion [156]. Five high-quality reference genomes were assembled, and a pan-genome of 770 Mb with 34,338 protein-coding genes was constructed, including 234 Mb and 6905 genes not covered by previous reference genomes [156]. Additionally, 35 candidate presence/absence variants (PAVs) associated with flowering time were identified, including *Phvul.003G185200*, a homolog of the *Arabidopsis* flowering gene *HDA5* [156].

In 2014, two independent domestication events in common beans were confirmed, originating from genetically distinct Mesoamerican and Andean gene pools that diverged before human colonization [33]. Less than 10% of the 74 Mb of sequence putatively involved in domestication was shared between these two events, highlighting the complexity of the domestication process [33].

In 2024, high genetic diversity and significant population differentiation between the Andean and Mesoamerican gene pools were concluded, which is crucial for identifying valuable traits through GWASs and for developing improved cultivars [157].

### 7.2. Functional Genomics and Gene Function

An ancient relaxation of an obligate short-day requirement in common beans occurred through the loss of *CONSTANS*-like gene function [74]. Additionally, a second major photoperiod sensitivity locus, *DTF4*, associated with the *CONSTANS*-like gene *COL2*, was identified. Distinct *col2* haplotypes were found to be associated with early flowering in Andean and Mesoamerican germplasm. This photoperiod adaptation can be fulfilled in two phases: initial reduction in sensitivity through *COL2* impairment and complete loss through *PHYA3*. *COL2* functions downstream of *PHYA3* to repress *FT* gene expression and may act in parallel with *PvE1* [73,74].

GWASs were performed to identify candidate genes and quantitative trait loci (QTLs) for early maturity [158], pod morphological traits and pod edibility [159], mineral content [160], chlorophyll content, photosynthetic activity, and flavonoid biosynthesis [161].

*PvPW1* on chromosome 06 was shown to be associated with pod width, and *PvPW1* was fine-mapped to a 42.3 kb region containing five genes. A SNP in *Phvul.006G072800* (encoding β-1,3-glucanase 9) was perfectly associated with pod width variation. Additionally, an InDel marker *PvM436* was developed for molecular breeding based on the polymorphism in *PvPW1* [75].

*MS-2* (*Phvul.003G031200*) was shown to cause male sterility due to a splice site mutation leading to a three-amino acid deletion, causing premature degradation of the tapetum. Exogenous application of IAA could increase the size and weight of seedless pods in *ms-2* mutants, with potential for processing or direct consumption [76].

*PvFtsH2* was identified as a critical factor in the degradation of photodamaged PSII RC D1 protein in common beans, demonstrating that *PvFtsH2* is essential for survival and maintaining photosynthetic activity by degrading photodamaged PSII RC D1 protein [77]. Genome-wide identification and a transcriptome analysis were conducted to disclose the function of GATA transcription factors on abiotic stress tolerance in common beans [162]. Upon evaluation of 255 yellow bean genotypes (YBC) against eight anthracnose races, a GWAS using 72,866 SNPs was conducted and major resistance loci mapped to several chromosomes, especially novel regions on Pv02, Pv05, and Pv07 [163]. Candidate R-genes with NB-ARC/LRR domains clustered on Pv02 and Pv04 [163]. Genetic fine-mapping of the anthracnose resistance allele *Co-1(4)* was performed in the Andean cultivar AND 277, localizing the resistance locus to a 40.51 kb region on chromosome Pv01 [164]. GWASs were performed to identify candidate genes and quantitative trait loci (QTLs) for drought tolerance and related traits [165,166], resistance to *Xanthomonas phaseoli* pv. *phaseoli* (Xpp) [167], and resistance to the rust pathogen [168]. Key *PvIQD* genes were shown to be consistently upregulated under salt stress, suggesting their potential use in breeding salt-tolerant varieties [169]. The *vsiRNA6163* targets and downregulates *PvTCP2*, activating the autophagy pathway and degrading *PvERD15*, reducing stomatal aperture. This mechanism was closely related to changes in soluble sugar content, and autophagy inhibitors could reverse the drought-tolerant phenotype [170].

## 8. Chickpea (*Cicer arietinum*)

Chickpeas (*Cicer arietinum*) are nutritious and drought-tolerant legumes known for their nutty flavor and buttery texture. These beige, round beans are high in protein, fiber, and vitamins, making them a staple in Mediterranean and Middle Eastern cuisines, particularly as the main ingredient in hummus and falafel.

### 8.1. Genome Sequencing and Assembly

The initial draft genome sequence of chickpeas was reported in 2013 [35], using next-generation sequencing platforms, bacterial artificial chromosome (BAC) end sequences, and a genetic map. The 520 Mb assembly covered 70% of the predicted 740 Mb genome length and included over 80% of the gene space. The genome analysis predicted the presence of 27,571 genes and identified approximately 210 Mb of repetitive elements within the genome [35].

In 2024, a super-pan-genome was constructed using high-quality genomes for eight wild annual species of the *Cicer* genus along with two previously published chickpea genomes [36]. The genome sizes varied across species, ranging from 435 Mb to 895 Mb, with gene counts from 23,486 to 29,642. A comparative analysis revealed potential ancient rearrangement events and identified three major gene pools within the *Cicer* genus [36]. A total of 678 structural variants (SVs) were found to overlap with 200 flowering time-related genes, with an average of 3.4 SV events per gene. Similarly, 1667 SVs were identified overlapping with 556 disease resistance (R) genes, averaging 3.0 SV events per R gene [36].

In 2021, a comprehensive pan-genome analysis was conducted by sequencing 3171 cultivated and 195 wild accessions, resulting in a pan-genome of 592.58 Mb with 29,870 genes [171]. A total of 124,833 SNPs (single nucleotide polymorphisms) were identified and associated with important traits such as seed weight and yield [171]. In addition, based on 3366 sequenced chickpea genomes, 1582 new genes were identified, and a comprehensive genetic variation map was constructed. Selection sweeps and deleterious mutations affecting cultivated lines’ adaptability were identified [172].

### 8.2. Functional Genomics and Gene Function

*MPL1*, a C2H2 zinc finger transcription factor, was proved to regulate leaflet initiation in chickpeas [78]. *MPL1* expression is complementary to *CaLFY*, a gene maintaining meristem activity, with *MPL1* inhibiting *CaLFY* to promote leaflet differentiation. Mutants lacking *MPL1* have high *CaLFY* activity, leading to the formation of multi-pinnate leaves with up to 40 leaflets [78].

Natural alleles and haplotypes of the *ABCC3* transporter gene associated with seed size and weight were identified [79]. High-seed-weight haplotypes are dominant in kabuli varieties, while desi and wild accessions have low-seed-weight haplotypes. *CaABCC3(6)* regulates seed weight by transporting glutathione conjugates and can be introgressed to improve desi varieties without compromising other agronomic traits [79].

Grain yield, photosynthetic activity, and molecular features under drought stress in 36 genotypes were assessed in 2024 [173]. Metabolites such as L-lipoic acid, fructose, and sorbitol involved in drought response during pod filling were identified. Genotypes were classified into four performance categories based on stress sensitivity indices. Higher flux in glycolysis and the MEP pathway were observed in drought-tolerant genotypes [173].

*Ca-miR164e* was found to target the C-terminal region of *CaNAC100*, a nuclear-localized transcription factor [80]. The overexpression of *CaNAC100* increases seed protein content (SPC) and seed weight but reduces seed number and yield. Conversely, the overexpression of *Ca-miR164e* downregulates *CaNAC100* and SSP transcripts, decreasing SPC, demonstrating that *CaNAC100* transactivates SSP-encoding genes by binding to their promoters [80].

Genome-wide identification and a transcriptome analysis were conducted to elucidate the function of the PHO1 family on Pi transport and root nodulation [174] and the function of the SPL gene family [175].

Based on a GWAS or QTLseq, the QTL region and candidate genes were identified for *Fusarium* wilt resistance [176]. A chickpea genomic resource of an integrated transcriptome was built to map the regulatory network of coding and long non-coding RNAs [177]. Through meta-analysis, extracellular matrix proteome, and phosphoproteome of chickpeas, novel proteoforms were revealed [177].

## 9. Cowpea (*Vigna unguiculata*)

Cowpeas (*Vigna unguiculata*) are versatile, nutritious, and drought-resistant legumes widely cultivated in tropical and subtropical regions. They are known for their green or yellow pods containing edible seeds and are high in protein and fiber.

The structure of the 11 chromosomes of cowpeas was revealed using molecular cytogenetics, providing a cytogenetic basis for understanding the genome organization of this species [178]. The nuclear genome size of cowpeas was estimated to be 640.6 Mb using cytometry [37]. The assembled genome of the single-haplotype inbred line IT97K-499-35 totals 519 Mb, with nearly half composed of repetitive elements, particularly Gypsy retrotransposons, which are enriched in recombination-poor pericentromeric regions [37]. Additionally, an inversion of 4.2 Mb was identified among landraces and cultivars, including a gene associated with interactions with the parasitic weed *Striga gesnerioides* [37].

In 2023, the first cowpea pan-genome of this species was constructed using de novo assemblies of six accessions, resulting in a mean genome size of 449.91 Mb [38]. The pan-genome includes 21,330 core genes and 23,531 non-core genes. Notably, presence/absence variants (PAVs) were found to contribute to traits such as black seed-coat color [38].

In 2024, high-quality genomes for two cowpea varieties, G98 (long-podded) and G323 (grain-type), were assembled with genome sizes of 568.24 Mb and 552.66 Mb, respectively [39]. In these genomes, 33,159 and 33,222 genes were annotated, respectively. A substantial portion of the genome was identified as repetitive sequences, primarily transposable elements, revealing significant genetic differentiation between the two subspecies, with distinct patterns of gene family expansion and contraction [39].

In 2019, CRISPR/Cas9-mediated genome editing technology was adapted to efficiently disrupt a representative symbiotic nitrogen fixation (SNF) gene in *Vigna unguiculata*, demonstrating the potential of genome editing for improving agronomic traits in cowpeas [179]. Genome-wide identification and a transcriptome analysis were conducted to elucidate the function of the *P5CR* and *αTPS6* genes on water deficit and high temperatures [180] and Thaumatin-like Proteins on both biotic and abiotic stresses [181].

Based on a GWAS and QTLseq, the QTL region or candidate genes were identified for seed protein content [182] and snake-like pod surface [183].

## 10. Pigeonpea (*Cajanus cajan*)

Pigeonpeas (*Cajanus cajan*) are important legume crops, particularly in arid and semi-arid regions, known for their drought tolerance and high nutritional value. They are a staple food in many developing countries and play a crucial role in sustainable agriculture due to their ability to fix nitrogen and improve soil fertility.

In 2012, the first draft genome sequence of pigeonpeas (*Cajanus cajan*) was released with 237.2 Gb of sequence data, which was assembled into scaffolds representing 72.7% (605.78 Mb) of the 833.07 Mb genome [40]. The genome analysis predicted 48,680 genes and highlighted the potential role of gene families related to drought tolerance in the domestication and evolution of pigeonpeas [40].

In 2020, a new refined version of the genome assemblies was presented, facilitating more accurate annotations and better resolution of complex genomic regions [184].

The pan-genome of pigeonpeas was constructed using 89 accessions, resulting in a pan-genome size of 622 Mb with 48,067 core genes and 7445 accessory genes [41]. A functional analysis of variable genes suggested roles in proteolysis and signal transduction. Phenotypic diversity in traits such as seed weight and flowering time were associated with presence–absence variations (PAVs) [41]. Several miRNAs with potential roles in stress response and developmental processes were identified using a genome-wide identification of miRNAs in pigeonpeas [185]. A genomic survey of high-throughput RNA-Seq data demonstrated long intergenic non-coding RNAs (lincRNAs) is associated with cytoplasmic male-sterility and fertility restoration in pigeonpeas [186].

Genome-wide identification and a transcriptome analysis were performed to investigate the function of P-type ATPases on pollen fertility and development [187]; MYB family genes on drought stress [188]; GRAS gene family on tissue development and stress responses [189]; and cytokinin oxidase/dehydrogenase (CKX) gene on seed number [190]. In addition, a physical map of lncRNAs and lincRNAs linked with stress-responsive miRs was disclosed [191].

## 11. Mung Beans (*Vigna radiata*)

Mung beans (*Vigna radiata*) are small, green legumes that are highly nutritious and rich in protein, fiber, and vitamins. They are widely cultivated and can be consumed whole, sprouted, or made into flour, and they are used in various cooking applications.

The initial efforts in mung bean genomics were marked by the generation of 470,024 genome shotgun sequences covering 100.5 Mb of the mung bean genome [42]. The first draft genome sequence of mung beans was published in 2014, covering approximately 75% of the estimated 484 Mb genome size [192]. This assembly included 27,114 predicted protein-coding genes [192]. A near-complete genome sequence provided higher resolution for gene annotation and the identification of regulatory elements [193]. A high-quality reference genome for the mung bean variety “JL7,” covering 99.13% of the estimated genome size (475.19 Mb) with a contig N50 length of 10.34 Mb, was reported [43]. The genome contained 53.45% repetitive sequences, primarily LTR retrotransposons (33.05%), and a total of 40,125 protein-coding genes were annotated [43]. The high-quality, gap-free T2T genome for the variety “Weilü No. 9” was achieved in 2024 [44]. The assembled genome size is 500 Mb, with 11 chromosomes and a contig N50 length of 46 Mb, containing 49.17% repetitive sequences. A total of 28,320 protein-coding genes were annotated.

The chloroplast genome sequence of mung beans was reported in 2009, providing insights into the structural organization and phylogenetic relationships within the legume family [194].

Pan-genome studies were conducted in 2022, facilitating the understanding of genetic diversity and identifying genes associated with important agronomic traits [43].

Genome-wide identification and a transcriptome analysis were performed to investigate the function of PEBP family genes on photoperiod [195]; cytochrome c oxidase-coding gene on cadmium stress [196]; JmjC domain-containing gene family [197] and bZIP members [198] on abiotic stress; and *VrNAC15* on drought resistance [199].

## 12. Adzuki Beans (*Vigna angularis*)

Adzuki beans (*Vigna angularis*) are small, reddish-brown legumes native to East Asia, known for their sweet, nutty flavor and high protein and fiber content. They are widely used in Asian cuisine.

The first draft genome of adzuki beans was released in 2015, covering 75% of the estimated genome and mapped to 11 pseudochromosomes. The genome assembly included 26,857 high-confidence protein-coding genes, identified through RNA-seq analysis of different tissues [200]. Additionally, a high-quality draft genome sequence of adzuki beans was released through whole-genome shotgun sequencing, covering 83% of the genome (450 Mb) with an N50 of 38 kb, in which a total of 34,183 protein-coding genes were predicted [201].

Furthermore, a high-quality chromosome-level reference genome of adzuki beans was released along with the resequencing of 322 accessions, including wild and cultivated varieties [45]. The reference genome of the cultivated variety “Jingnong6” was assembled using a combination of Illumina, PacBio, and Hi-C technologies, resulting in a genome size of 489.8 Mb with a contig N50 of 16.1 Mb and a scaffold N50 of 41.6 Mb, with 97.8% of the data anchored to 11 chromosomes [45]. Comparative genomics with 11 other species revealed 16,641 gene families, with 846 orthogroups unique to adzuki beans [45]. Meanwhile, key genes associated with domestication traits, such as pod dehiscence, yield, and seed coat color, were identified through GWASs and selective sweep analyses. For example, a nonsynonymous SNP in the gene *VaCycA3;1* was associated with seed number and growth habit, while *VaANR1* was linked to seed coat color [45]. A population analysis of 50 accessions (including wild, semiwild, landraces, and improved varieties) detected strong selection signals in domestication. The wild adzuki beans might be a preliminary landrace in the domestication process [201].

Under low nitrogen supply conditions, exogenous calcium can promote seedling growth in adzuki beans through the regulation of nitrogen metabolism [202].

## 13. Rice Bean (*Vigna umbellata*)

*Vigna umbellata*, commonly known as rice beans, is a trailing plant with small, green pods and white or purple flowers. It is native to Southeast Asia and is drought tolerant, often grown for food and fodder.

A high-quality genome assembly of rice beans was reported, with a size of 525.6 Mb and 26,736 high-confidence protein-coding genes annotated [46]. Based on resequencing of 440 landraces, rice beans were classified into three genetic groups: South Asia and Southeast Asia, Southern China, and Northern China. The South Asia and Southeast Asia group exhibited the highest genetic diversity, supporting the hypothesis that this region is the origin of rice beans [46].

*PRR3b*, *FT*, *FUL*, and *CYP78A6* were proposed to be key loci or candidate genes related to photoperiod sensitivity, flowering time, and yield-related traits [46]. A 2 nt deletion in the *TFL1* gene was found to be associated with increased yield in certain landraces from Southern China [46]. Based on comparative transcriptomic analysis, *VuNAR1* was revealed to moderate Al resistance by regulating cell wall pectin metabolism via directly binding to the promoter of *WAK1* and inducing its expression [203].

## 14. White Lupin

White lupin (*Lupinus albus*) is a significant grain legume crop, highly valued for its rich protein content, adaptability to poor soils, and potential to contribute to sustainable agriculture.

A high-quality genome sequence of a cultivated accession of white lupin (2n = 50, 451 Mb) was reported, along with de novo assemblies of a landrace and a wild relative using long-read sequencing technologies [47]. The first pan-genome of white lupin was constructed using 39 accessions, resulting in a pan-genome size of 32,068 core genes, enabling the identification of selection sweeps related to low-alkaloid content [204]. Additionally, the assembly of the white lupin genome enabled researchers to identify that the PUE (phosphorus-use efficiency) genes in white lupin have been expanded through WGT (whole-genome triplication) as well as tandem and dispersed duplications [205]. Four main pathways for high PUE were characterized, including carbon fixation, cluster root formation, soil-P remobilization, and cellular-P reuse. Moreover, auxin modulation may be important for cluster root formation through the involvement of potential genes *LaABCG36s* and *LaABCG37s* [205]. A high-density consensus linkage map of white lupin was developed to map important agronomic traits such as vernalization requirement, seed alkaloid content, and resistance to anthracnose and Phomopsis stem blight [206]. The high-efficiency CRISPR/Cas9-based multiplex genome editing system was successfully adapted in white lupin using endogenous promoters with a *Lalb_Chr05g0223881* gene, encoding a putative trehalase [207]. Furthermore, the trehalase gene *LaTRE1* was proven to regulate the formation of cluster root, a specialized root structure, and the promotion of organic acid production under phosphorus deficiency [208].

## 15. *Mimosa*

*Mimosa* is a genus of plants known for their delicate, feathery leaves and pink, pom-pom-like flowers. Many species, like *Mimosa pudica*, have touch-sensitive leaves that fold inward when disturbed [209].

A high-quality, haplotype-resolved reference genome of *Mimosa bimucronata* was assembled, covering 648 Mb and anchored in 13 pseudochromosomes [48]. A total of 32,146 protein-coding genes were annotated, with 31,035 genes in haplotype A and 31,440 in haplotype B [48]. Two whole-genome duplication (WGD) events, occurring approximately 2.9 and 73.5 million years ago, were inferred. Transcriptome and co-expression network analyses highlighted the involvement of aquaporins (AQPs) and Ca^2+^-related ion channel genes in leaf movement. Additionally, nodulation-related genes were identified, with a focus on the structure and evolution of the key gene *NIN* involved in symbiotic nitrogen fixation (SNF) [48].

The complete chloroplast genome sequences of *Mimosa diplotricha* and its variant *Mimosa diplotricha* var. *inermis* were reported [210]. These invasive species, introduced to China in the 19th century, pose significant ecological threats. The chloroplast genome of *M. diplotricha* is 164,450 bp long, while that of *M. diplotricha* var. *inermis* is 164,445 bp long [211]. A total of 84 genes were annotated, including 54 protein-coding genes, 29 tRNA genes, and 1 rRNA gene [211]. Phylogenetic analysis based on the chloroplast genomes of 22 related species showed that *M. diplotricha* var. *inermis* is most closely related to *M. diplotricha*, and this clade is sister to *Mimosa pudica*, *Parkia javanica*, *Faidherbia albida*, and *Acacia punctulata* [211]. Functional studies on the chloroplast genome benefit invasion risk monitoring of these species [210,211].

## 16. *Dalbergia*

*Dalbergia*, commonly known as rosewood, is highly valued for its dense, richly colored wood, which is often used in high-quality furniture, musical instruments, and luxury crafts due to its slow growth and the wood’s aromatic scent and durability.

The first chromosome-level draft genome of *Dalbergia odorifera*, a species highly valued for its medicinal and commercial properties, was reported [49]. Approximately 97.68% of the genome was assembled, which is approximately 653.45 Mb in size. The genome contains 30,310 predicted protein-coding genes, with 92.6% functionally annotated. Repetitive elements account for 54.17% of the genome [49].

The chloroplast genomes of nine *Dalbergia* species were sequenced and compared, revealing high conservation in terms of size, structure, and gene content [212]. The chloroplast genomes exhibited low sequence divergence, indicating a relatively stable evolutionary history within the genus. Eight mutation hotspots, including six intergenic spacer regions and two coding regions, were identified and proposed as candidate DNA barcodes for species identification [212]. Additional chloroplast genome studies on *Dalbergia latifolia* and *Dalbergia oliveri* further confirmed the conservation of chloroplast genome structure within the genus and identified additional variable regions that could be used as markers for phylogenetic and conservation studies [213,214].

Additionally, a comparison of 35 complete chloroplast genomes from the genus *Dalbergia* identified barcode regions for identifying wood of *Dalbergia odorifera* and *Dalbergia tonkinensis* [215]. The possible involvement of R2R3-MYB TFs in heartwood formation [216] and microRNAs in xylem development [217] in *Dalbergia odorifera* was disclosed through a genome-wide survey.

## 17. Licorice (*Glycyrrhiza* spp.)

The draft genome assembly of *Glycyrrhiza uralensis*, a medicinal legume whose roots are widely used in herbal medicine, was first reported [50]. The genome assembly, approximately 379 Mb in size, contains 34,445 predicted protein-coding genes. Gene clusters involved in isoflavonoid biosynthesis, including 2-hydroxyisoflavanone synthase (*CYP93C*), *HI4OMT*, and *7-IOMT*, were identified and showed conserved microsynteny with related legumes. The genome annotation also predicted genes in the P450 and UDP-dependent glycosyltransferase (UGT) superfamilies, which are involved in triterpenoid saponin biosynthesis [50].

A comprehensive genomic study on *Glycyrrhiza uralensis* was conducted, which included whole-genome re-sequencing of 60 samples from diverse habitats. A total of 6,985,987 SNPs were identified, and 117,970 high-quality SNPs were filtered, suggesting that *G. uralensis* has adapted to different environments through selective pressures on specific metabolic pathways [218]. UDP-glucose dehydrogenase (UGD) isoforms in *Glycyrrhiza uralensis* were characterized [219]. UGD plays a crucial role in the biosynthesis of bioactive compounds, including triterpenoid saponins, in licorice, which can be targeted for genetic improvement [219].

The draft genome assembly of *G. uralensis* revealed gene clusters involved in isoflavonoid biosynthesis, such as *CYP93C*, *HI4OMT*, and *7-IOMT*, which are critical for the production of bioactive isoflavonoids [50]. The synthesis of 11-oxo-β-amyrin and glycyrrhetinic acid in *Saccharomyces cerevisiae* is boosted by pairing a novel oxidation and reduction system from legume plants [220].

Glycyrrhizin, the most important bioactive compound, determines the quality of medicinal licorices. With the aid of the constructed regulatory network of glycyrrhizin in *G. inflata* based on the assembled genomes and transcriptomes, *GibHLH9*, *GibHLH53*, and *GibHLH174* were identified as key transcription factors that promote glycyrrhizin by transactivating the expression of *GiCSyGT* and *GiUGT73P12*, respectively [221].

## 18. Lentil (*Lens culinaris* Medik.)

Lentils (*Lens culinaris* Medik.) are globally important cool-season grain legumes that serve as a rich source of protein for human consumption.

Kaur et al. (2011) conducted pioneering work in lentil genomics by sequencing tissue-specific cDNA samples from six distinct lentil genotypes using Roche 454 GS-FLX Titanium technology [222]. Approximately 1.38 million expressed sequence tags (ESTs) were generated, which were assembled into 15,354 contigs and 68,715 singletons. A comparative analysis with the genomes of *Medicago truncatula* and *Arabidopsis thaliana* identified 12,639 and 7476 unique matches, respectively [222]. A total of 25,592 lentil unigenes were annotated from GenBank. Additionally, 20,419 unique hits were observed when compared to the *Glycine max* genome, covering about 31% of the known gene space.

The lentil pan-transcriptome was estimated to consist of 15,910 core genes and 24,226 accessory genes [223]. The complete chloroplast genome of *Lens tomentosus* is 123,548 bp long and contains 107 genes [224]. A phylogenetic analysis clustered the accessions into four groups, with *Lens nigricans* showing the greatest allelic differentiation when the genetic and allelic diversity among 467 wild and cultivated lentil accessions from diverse regions were analyzed [225].

A total of 7494 differentially expressed genes were identified to be related to temperature stresses in lentils [226]. Novel genes associated with herbicide tolerance were identified through a Meta-GWAS analysis [227]. *LcELF3a* was found to be substantially involved in promoting lateness using the whole-genome resequencing (WGRS)-based QTL-seq approach [228]. When genomic selection was applied to 2081 lentil breeding lines to predict breeding values for traits such as grain yield, disease resistance, stress tolerance, and seed quality, genomic estimated breeding values (GEBVs) had higher accuracy than BLUP estimated breeding values (EBVs) using 64,781 SNPs [229].

## 19. Comparative Genomic Studies Between Legume Species

Given the substantial genetic diversity among legume species, comparative genomics is generally performed to aid genome assembly through syntenic analysis and to elucidate domestication history and potential for crop improvement. Additionally, a comparative analysis with other species can greatly facilitate the identification of key genes and pathways associated with important agronomic traits. For instance, Basso et al. (2024) compared two chickpea and two lentil cultivars with contrasting branching patterns and identified 10 chickpea genes and 7 lentil genes that play key roles in regulating branching [230]. A functional analysis of adzuki beans has indicated that differences in starch and fat content between adzuki beans and soybeans are primarily due to transcriptional abundance rather than copy number variations of genes related to starch and oil synthesis [200]. A comparative analysis of the chloroplast genomes of *Cajanus cajan* and its wild relative *Cajanus scarabaeoides* revealed conserved genome structure and gene content, providing insights into the evolutionary history and domestication of pigeonpeas [231]. A comprehensive phylogenetic analysis, placing cowpeas within the context of other related legumes, enabled the identification of key genes associated with important agronomic traits, such as yield, quality, and resilience through GWASs [39].

In addition to these comparative genomic studies, functional genomic approaches have also yielded significant results. A total of 21 potential drought-responsive NAC genes were identified in chickpeas, pigeonpeas, and groundnuts through genome-wide analysis and expression studies, among which correlation between expression, transcriptional regulation, and drought tolerance were observed, and these genes can be served as valuable resources for breeding drought-tolerant legume crops [232]. Putative targets of the *VEGETATIVE1/FULc* gene in peas, crucial for compound inflorescence development, were identified in legumes. Using transcriptome comparison and VIGS, *PsHUP54*, a gene that controls I2 meristem activity and significantly increases pod size and seed yield, was revealed, demonstrating a promising tool for improving yield in peas and other legumes [233]. Multi-omic approaches were used to identify novel aphid effector candidates associated with virulence and avirulence phenotypes in interactions with *Medicago truncatula*. Differential gene expression and proteomics revealed key proteins, including aminopeptidase N, linked to these phenotypes [234].

Moreover, the potential of genomic selection, speed breeding, and CRISPR/Cas9 genome editing to develop phosphorus-use efficiency (PUE)-efficient cultivars, reducing phosphorus fertilizer overuse and promoting sustainable agriculture, was reviewed [235]. The review highlighted the use of QTL mapping, genome-wide association studies, and advances in genomic resources to uncover genomic regions related to PUE. Functional genomics, metabolomics, and proteomics have identified key genes and pathways involved in PUE.

## 20. Future Perspectives

The Fabaceae family is highly diverse, encompassing a wide variety of species, particularly wild relatives. Despite sharing many similarities, leguminous plants exhibit rich diversity in various aspects, including growth habits (trailing or erect), stem tissue composition and structure (herbaceous or woody), seed size, variations in oil and protein content, secondary metabolites, resistance to biotic and abiotic stresses, and nitrogen fixation efficiency.

Currently, many leguminous species still lack high-quality reference genomes. The availability of one or more high-quality genomes is mostly limited to important crops or model plants used in production and research, such as soybeans, common beans, and *Medicago*. However, as sequencing technology and assembly software continue to advance, most leguminous species will gradually acquire one or more high-quality reference genomes (Table 1).

Comparative and in-depth functional genomics can help efficiently interpret the morphological and metabolic differences among these plants. This approach will facilitate the comprehensive utilization and improvement of leguminous species. In summary, each leguminous plant has a unique genomic composition and structural characteristics. Different genes regulate the distinct nutritional components, physiologically active substances, or secondary metabolites of leguminous plants in humans or animals through various metabolic pathways.

Furthermore, the availability of extensive collections of *Tnt1*-tagged and fast neutron deletion mutants in species such as *M. truncatula* and soybeans, along with efficient mutant screening methods, is advancing legume functional genomics. These resources support gene cloning, functional studies, and the dissection of gene regulatory networks, thereby accelerating progress in legume genetics and breeding. Based on accumulated research findings across species, comparative genomics can further clarify the commonalities and differences among leguminous plants at the genomic and functional genomic levels.

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
