# Peer review of "Functional Genomics: From Soybean to Legume"

_ijms, 2025, doi:10.3390/ijms26136323_

Round 1
Reviewer 1 Report
Comments and Suggestions for Authors
The authors conducted a review titled Functional Genomics: From Soybean to Legume, which is highly meaningful and also challenging. However, the organization of this review suffers from major problems and requires substantial revision. The specific issues are as follows, but are not limited to the questions listed below.
The abstract lacks content. It ends with “This review aims to summarize the current status of functional genomics research in several major important leguminous species.” From the abstract alone, readers cannot even tell what specific focus the review takes. At the very least, the authors should summarize the content and innovation points of this review.
“Leguminous crops can be categorized into staple grains, oil crops, vegetables, and miscellaneous grains. Soybeans and peanuts are primarily used as staple grains or oil crops, while cowpeas, peas, and common beans are generally consumed as vegetables.” The authors even omitted the forage function of leguminous plants here, which is a very important function of legumes.
In the section Genome Sequencing and Assembly, the first paragraph discusses resequencing, but then suddenly, in the next part starting with “Meanwhile,” it abruptly shifts to reference genome de novo assembly. These are completely different concepts, and the transition lacks logical flow. Meanwhile, the following paragraph “The quality of the soybean genome has been significantly improved” is even written entirely in italics. Also, the context lacks any logical coherence.
“The genome was assembled using Pacific BioSciences HiFi reads and integrated into chromosomes using HiC.” The use of “integrated” is incorrect; HiC is used to assist in scaffold ordering and orientation to produce chromosome-level assemblies. At the same time, please verify the content. For a genome like that of soybean, PacBio and HiC alone are insufficient for T2T assembly. Please check whether ultra-long ONT is missing.
When presenting the results of genome assembly, multiple versions (ZH13, ZJ13, Jack, etc.) are listed side by side, but without systematic comparison or summary evaluation. These should be integrated and compared, rather than displayed in a fragmented way.
PDH1 [21], Gene names should be italicized.
“Resequencing efforts have also contributed significantly to our understanding of soybean genetics. A high-quality genome of Nongdadou2 (NDD2) was constructed, and 547 accessions were resequenced, identifying numerous novel structural variations (SVs) and genes [24].” This section lacks substantial content.
In the section Functional Genomics and Gene Function, there are numerous studies on gene function. However, what is needed is the establishment of functional annotation systems, rather than scattered single-gene validations. Many of the cited contents are phenotypic and functional studies of individual genes, mostly based on traditional functional validation approaches. These do not fall under the scope of systematic functional genomics. In essence, they are “classical gene function studies” rather than “functional genomics.” Topics such as flowering regulation, plant height, pod setting, disease resistance, and male sterility are piled up in a fragmented manner, without structural organization based on functional pathways, regulatory networks, or signaling modules. I believe this part needs major restructuring, with a focus on omics-driven gene discovery to match the topic of the review.
The LjNIN gene, which encodes a transcription factor, was identified as playing a crucial role in nodule development and nitrogen fixation [43].
Similarly, the LjSymRK gene, involved in the early stages of rhizobial infection, was identified and functionally validated through mutant analysis and complementation studies [43]. What does this have to do with functional genomics? From the description, it is merely conventional molecular biology function identification, which is extremely common. This content is almost unrelated to functional genomics.
Similarly, the paragraphing here is highly unreasonable. For example, “the potential regulatory roles of WRKY transcription factors in flavonoid metabolism during nodule formation were disclosed” contains no substantial information at all.
A pan-genome was also established from 116 pea accessions, identifying unique pan-genes in wild species such as Pisum abyssinicum and P. fulvum, which are associated with disease resistance and development [65]. This sentence is very vague and lacks substantive information. What kind of disease resistance is being referred to—fungal, bacterial, viral, or broad-spectrum? What does “development” mean? Does it refer to flower development, seed development, root architecture, or overall development? How many unique pan-genes were identified? What is the core message of the article?
Multiple significant SNPs and candidate genes for iron, phosphorus, and zinc were identified by GWAS, which can be used for genetic improvement in pea for nutritional traits [68]. Lacks actual content.
Much of the content related to Pisum sativum lacks a logical narrative thread.
The diploid progenitors of peanut, Arachis duranensis and Arachis ipaensis, have also been sequenced, providing insights into the ancestral origins of the cultivated peanut [76]. What useful information can readers actually obtain from this sentence?
How is AhSAMS1 related to functional genomics?
High-throughput sequencing technologies have been used to analyze genetic diversity in peanut populations [82], while molecular markers have been developed and used in marker-assisted selection (MAS) programs to accelerate the breeding of improved peanut varieties [83]. What is the logical link of this content to the previous discussion?
Ambachew et al. (2024) revealed high genetic diversity and significant population differentiation between the Andean and Mesoamerican gene pools, which is crucial for identifying valuable traits through GWAS and for developing improved cultivars [92]. What is the actual contribution of this article?
A GWAS was conducted to identify loci associated with anthracnose resistance in the Yellow Bean Collection [93]. Again, what concrete content is being conveyed here?
GWAS were performed to identify candidate genes and quantitative trait loci (QTLs) for drought tolerance and related traits [95,96], for mineral content [97], and for chlorophyll content, photosynthetic activity, and flavonoid biosynthesis [98]. Again, what actual findings are presented here?
Several stress-responsive genes were identified through RNA-Seq analysis in the initial draft genome sequence by Jain et al. (2013) [106].
MPL1, a C2H2 zinc finger transcription factor regulating leaflet initiation in chickpea, was identified by Liu et al. (2023) [110]. What is the substance of [106]? This sentence contains no meaningful information.
The excessive number of “xx et al. 20xx” citations severely affects the logical coherence of the entire manuscript. The authors cite many papers but fail to synthesize them; much of the content is fragmented. The biggest problem is the lack of substantive information—many statements do not offer readers any practical knowledge. In particular, the Functional Genomics and Gene Function section includes a lot of content that has no relevance to functional genomics. If the focus is merely on gene function, many results from molecular validation will lead to significant bias. The authors are advised to center the discussion on genes, gene clusters, and candidates identified through functional genomics approaches. Rather than expanding the volume of refs, the review should deepen the content. The current structure is a disorganized compilation of vague, unsubstantial refs and lacks both logic and depth.
Comments on the Quality of English Language
The author's logic and overall organization of the manuscript are extremely poor.
Author Response
Comments 1:
The abstract lacks content. It ends with “This review aims to summarize the current status of functional genomics research in several major important leguminous species.” From the abstract alone, readers cannot even tell what specific focus the review takes. At the very least, the authors should summarize the content and innovation points of this review.
Response 1: Thank you for your comment. Given that the topic is broad and encompasses extensive information, it is indeed necessary to summarize the specific focus or interests. Accordingly, we have restructured the Abstract.
Comments 2:
“Leguminous crops can be categorized into staple grains, oil crops, vegetables, and miscellaneous grains. Soybeans and peanuts are primarily used as staple grains or oil crops, while cowpeas, peas, and common beans are generally consumed as vegetables.” The authors even omitted the forage function of leguminous plants here, which is a very important function of legumes.
Response 2: Thank you for your clear correction, we have revised accordingly throughout the manuscripts
Comments 3:
In the section Genome Sequencing and Assembly, the first paragraph discusses resequencing, but then suddenly, in the next part starting with “Meanwhile,” it abruptly shifts to reference genome de novo assembly. These are completely different concepts, and the transition lacks logical flow. Meanwhile, the following paragraph “The quality of the soybean genome has been significantly improved” is even written entirely in italics. Also, the context lacks any logical coherence.
Response 3: Thank you for your point out the weakness of the writing, we have revised accordingly and tried to make the expression currently and logically. Also we have been carefully through the manuscript and hope there are no such mistakes anymore.
Comments 4:
“The genome was assembled using Pacific BioSciences HiFi reads and integrated into chromosomes using HiC.” The use of “integrated” is incorrect; HiC is used to assist in scaffold ordering and orientation to produce chromosome-level assemblies. At the same time, please verify the content. For a genome like that of soybean, PacBio and HiC alone are insufficient for T2T assembly. Please check whether ultra-long ONT is missing.
Response 4: Thank you very much for your correction, we have organized all contents regarding genome sequencing and assembly, hope in the new version, all description regarding the genome sequencing and assembly are properly described.
Comments 5:
When presenting the results of genome assembly, multiple versions (ZH13, ZJ13, Jack, etc.) are listed side by side, but without systematic comparison or summary evaluation. These should be integrated and compared, rather than displayed in a fragmented way.
Response 5: Agree. we did not compare or analyze both results, now we compared the two assembly versions of ZH13 as well as other cultivars.
Comments 6:
PDH1 [21], Gene names should be italicized.
Response 6: Thank you for your point out, we have checked all gene names and protein names and make sure the gene names are all italicized.
Comments 7:
“Resequencing efforts have also contributed significantly to our understanding of soybean genetics. A high-quality genome of Nongdadou2 (NDD2) was constructed, and 547 accessions were resequenced, identifying numerous novel structural variations (SVs) and genes [24].” This section lacks substantial content.
Response 7: Agree, we have revised this paragraph and all similar paragraphs.
Comments 8:
In the section Functional Genomics and Gene Function, there are numerous studies on gene function. However, what is needed is the establishment of functional annotation systems, rather than scattered single-gene validations. Many of the cited contents are phenotypic and functional studies of individual genes, mostly based on traditional functional validation approaches. These do not fall under the scope of systematic functional genomics. In essence, they are “classical gene function studies” rather than “functional genomics.” Topics such as flowering regulation, plant height, pod setting, disease resistance, and male sterility are piled up in a fragmented manner, without structural organization based on functional pathways, regulatory networks, or signaling modules. I believe this part needs major restructuring, with a focus on omics-driven gene discovery to match the topic of the review.
Response 8: Thank you for your critical comment, which has highlighted the weaknesses in our previous version. Accordingly, we have reorganized the entire content to ensure logical coherence and flow. Gene cloning has significantly benefited from the availability of reference genome sequencing, a trend that was already evident when E1 was cloned in 2012.
Regarding the functional genomics and gene function sections, we have structured the content to follow the chronological sequence of plant growth and development. This includes:
Flowering regulation, Vegetative growth processes (e.g., stem, leaf, and pod development),Nodulation mechanisms,Reproductive sterility phenomena, Resistance responses to abiotic and biotic stresses
Each section is presented in a hierarchical order to emphasize the interconnections between genetic functions and physiological processes.
Comments 9:
The LjNIN gene, which encodes a transcription factor, was identified as playing a crucial role in nodule development and nitrogen fixation [43].
Similarly, the LjSymRK gene, involved in the early stages of rhizobial infection, was identified and functionally validated through mutant analysis and complementation studies [43]. What does this have to do with functional genomics? From the description, it is merely conventional molecular biology function identification, which is extremely common. This content is almost unrelated to functional genomics.
Response 9: Indeed, while these genes are not directly related to functional genomics, their cloning and functional validation were significantly facilitated by the genome sequence published in 2008. We have revised the content accordingly to clarify their relevance to functional genomics.
Throughout the manuscript, we have retained key genes while removing those unrelated to functional genomics, such as LjNIN and LjSymRK. This adjustment ensures the content aligns more closely with the focus on functional genomics and maintains the logical coherence of the study.
Comments 10:
Similarly, the paragraphing here is highly unreasonable. For example, “the potential regulatory roles of WRKY transcription factors in flavonoid metabolism during nodule formation were disclosed” contains no substantial information at all.
A pan-genome was also established from 116 pea accessions, identifying unique pan-genes in wild species such as Pisum abyssinicum and P. fulvum, which are associated with disease resistance and development [65]. This sentence is very vague and lacks substantive information. What kind of disease resistance is being referred to—fungal, bacterial, viral, or broad-spectrum? What does “development” mean? Does it refer to flower development, seed development, root architecture, or overall development? How many unique pan-genes were identified? What is the core message of the article?
Response 10: Thanks for your indication. We have reorganized the relevant parts again and made substantial changes per your suggestion. In addition, we also added some newly reports.
Comments 11:
Multiple significant SNPs and candidate genes for iron, phosphorus, and zinc were identified by GWAS, which can be used for genetic improvement in pea for nutritional traits [68]. Lacks actual content.
Response 11:
We have revise accordingly and hoping the new version is acceptable.
Comments 12:
Much of the content related to Pisum sativum lacks a logical narrative thread.
Response 12: Agree. We reorganized this part and make it more logic and coherent.
Comments 13:
The diploid progenitors of peanut, Arachis duranensis and Arachis ipaensis, have also been sequenced, providing insights into the ancestral origins of the cultivated peanut [76]. What useful information can readers actually obtain from this sentence?
Response 13:
We reorganized this part.
Comments 14:
How is AhSAMS1 related to functional genomics?
Response 14:
We deleted this gene per your suggestion.
Comments 15:
High-throughput sequencing technologies have been used to analyze genetic diversity in peanut populations [82], while molecular markers have been developed and used in marker-assisted selection (MAS) programs to accelerate the breeding of improved peanut varieties [83]. What is the logical link of this content to the previous discussion?
Response 15:
We have reorganized this part per suggestion, hoping the revised version is logic and coherent.
Comments 16:
Ambachew et al. (2024) revealed high genetic diversity and significant population differentiation between the Andean and Mesoamerican gene pools, which is crucial for identifying valuable traits through GWAS and for developing improved cultivars [92]. What is the actual contribution of this article?
Response 16: agree.We have revised accordingly.
Comments 17:
A GWAS was conducted to identify loci associated with anthracnose resistance in the Yellow Bean Collection [93]. Again, what concrete content is being conveyed here?
Response 17: We have revised accordingly.
Comments 18:
GWAS were performed to identify candidate genes and quantitative trait loci (QTLs) for drought tolerance and related traits [95,96], for mineral content [97], and for chlorophyll content, photosynthetic activity, and flavonoid biosynthesis [98]. Again, what actual findings are presented here?
Response 18: Indeed, there are might be no much detail information disclosed due to the limited length of this paper, however, we think it is necessary for us to keep it to present a clue for readers’ further investigation.
Comments 19:
Several stress-responsive genes were identified through RNA-Seq analysis in the initial draft genome sequence by Jain et al. (2013) [106].
MPL1, a C2H2 zinc finger transcription factor regulating leaflet initiation in chickpea, was identified by Liu et al. (2023) [110]. What is the substance of [106]? This sentence contains no meaningful information.
Response 19: Thank you for your correction, we have reorganized and make it more logic and coherent.
Comments 20:
The excessive number of “xx et al. 20xx” citations severely affects the logical coherence of the entire manuscript. The authors cite many papers but fail to synthesize them; much of the content is fragmented. The biggest problem is the lack of substantive information—many statements do not offer readers any practical knowledge. In particular, the Functional Genomics and Gene Function section includes a lot of content that has no relevance to functional genomics. If the focus is merely on gene function, many results from molecular validation will lead to significant bias. The authors are advised to center the discussion on genes, gene clusters, and candidates identified through functional genomics approaches. Rather than expanding the volume of refs, the review should deepen the content. The current structure is a disorganized compilation of vague, unsubstantial refs and lacks both logic and depth.
Response 20: We have reorganized the whole manuscript e.g. we deleted almost all“xx et al. 20xx”. Due to the limited length, description of functions of some genes might be short, however, we hope we can keep these important genes and leave a cure for reader’s further investigation.

Reviewer 2 Report
Comments and Suggestions for Authors
This review is a comprehensive one that describes the progress of some legume genome analysis and the genes involved in important traits of each legume crop, across the legume family. This review is important for understanding the scientific development of the entire legume family. In this sense, I believe that this review also has significant scientific value.
However, the review text is just a list of information, and I could not understand the characteristics and problems peculiar to Fabaceae that are not found in other plants.It is fine as it is(no change is OK), but since this is a review of the Fabaceae, it would be good to have a part that discusses traits specific to the Fabaceae.
Minor revision
1. In table1, is it necessary to separate Pea [Cameor] and othertwo cultivars(ZW6,ZhewanNo1) with a ruled line?
2. In table2, there are three Chickpea followed by Chickpea. Also, is part of Soybean in bold?
Author Response
Comments 1:
" This review is a comprehensive one that describes the progress of some
legume genome analysis and the genes involved in important traits of
each legume crop, across the legume family. This review is important for
understanding the scientific development of the entire legume family. In
this sense, I believe that this review also has significant scientific
value.
However, the review text is just a list of information, and I could not
understand the characteristics and problems peculiar to Fabaceae that
are not found in other plants.It is fine as it is(no change is OK), but
since this is a review of the Fabaceae, it would be good to have a part
that discusses traits specific to the Fabaceae.
Response 1:
Thank you for your insightful feedback. As is well established, symbiotic signaling and developmental plasticity are hallmark traits of legumes. We have incorporated key legume-specific aspects related to nodulation and developmental plasticity into the Abstract, Introduction, and Discussion sections. Given the extensive diversity of the Fabaceae family—including numerous wild relatives—we have balanced the need for conciseness with the requirement to highlight legume specificity, mindful of the manuscript’s length constraints.
In the revised manuscript, we have restructured the content to ensure logical flow and coherence, while preserving scientific rigor. We trust these revisions meet your expectations.
Comments 2:
Minor revision
- In table1, is it necessary to separate Pea [Cameor] and othertwo
cultivars(ZW6,ZhewanNo1) with a ruled line?
- In table2, there are three Chickpea followed by Chickpea. Also, is
part of Soybean in bold?"
Response 2: thank you for your kind correction. We revised all the errors indicated and also deleted some unreverent genes. By the way, we only enclosed the important or typical ones in the two tables.

Round 2
Reviewer 1 Report
Comments and Suggestions for Authors
The author has addressed my questions, and I have no further comments.